# Mathematical Analysis and Update of ADM1 Model for Biomethane Production by Anaerobic Digestion

Stefano Bertacchi [1]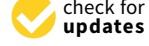, Mika Ruusunen [2], Aki Sorsa [2], Anu Sirviö [2] and Paola Branduardi [1,*]

[1] Department of Biotechnology and Biosciences, University of Milano-Bicocca, Piazza della Scienza 2, 20126 Milano, Italy; stefano.bertacchi@unimib.it

[2] Faculty of Technology, University of Oulu, Pentti Kaiteran katu 1 Linnanmaa, 90570 Oulu, Finland; mika.ruusunen@oulu.fi (M.R.); aki.sorsa@oulu.fi (A.S.); anu.sirvio@oulu.fi (A.S.)

\* Correspondence: paola.branduardi@unimib.it

**Abstract:** Biomethane is a renewable product that can directly substitute its fossil counterpart, although its synthesis from residual biomasses has some hurdles. Because of the complex nature of both biomasses and the microbial consortia involved, innovative approaches such as mathematical modeling can be deployed to support possible improvements. The goal of this study is two-fold, as we aimed to modify a part of the Anaerobic Digestion Model No. 1 (ADM1), describing biomethane production from activated sludge, matching with its actual microbial nature, and to use the model for identifying relevant parameters to improve biomethane production. Firstly, thermodynamic analysis was performed, highlighting the direct route from glucose to biomethane as the most favorable. Then, by using MATLAB® and Simulink Toolbox, we discovered that the model fails to predict the microbiological behavior of the system. The structure of the ADM1 model was then modified by adding substrate consumption yields in equations describing microbial growth, to better reflect the consortium behavior. The updated model was tested by modifying several parameters: the coefficient of decomposition was identified to increase biomethane production. Approaching mathematical models from a microbiological point of view can lead to further improvement of the models themselves. Furthermore, this work represents additional evidence of the importance of informatics tools, such as bioprocess simulations to foster biomethane role in bioeconomy.

**Keywords:** biomethane; anaerobic digestion; biorefinery; mathematical modeling; differential equations; bioeconomy

## 1. Introduction

At the end of the first decade of 21st century, the European Union (EU) Renewable Energy Directive 2009/28/EC required that by 2020, each member state had to rely on renewable energy for 20% of the total needs and 10% for transport alone [1]. Unfortunately, only Sweden and Austria have reached that target [2]; consequently, there is an urgent and binding need to push forward the use of alternatives to common fossil resources. Natural gas, namely methane ($CH_4$), is a prominent target to be substituted with renewable alternatives, as it is still widely used in the EU for industrial, domestic, and transport sectors. Its direct replacement is biomethane, which is the final product of anaerobic fermentation of organic matter, as part of the gaseous mixture called biogas. Biogas is in fact a blend of mainly $CH_4$ and $CO_2$: it can be deployed directly as a source of energy, but the presence of the already fully oxidized $CO_2$ reduces the overall calorific value. For this reason, several "upgrading" processes exist, aimed at purifying biomethane from biogas, to expand its use also to domestic and transport sectors via injection into the conventional pipelines [3,4].

Among the various residual biomasses used to match the low market value of biomethane [2,5], sewage sludges from wastewater treatments play a prominent role. They are side streams with a high environmental impact that can be reduced by further

manipulations, with a huge potential in terms of worldwide production of biogas, together with other biomasses, such as manure, organic fraction of municipal solid waste, and agricultural residues [6]. Since there has been increased growth of the European market of biomethane, led by Germany, several countries from both northern (e.g., Finland) and southern (e.g., Italy) Europe are interested in developing this sector, in order to fulfill the aforementioned goals of the EU over the coming years [2,7]. To make the production profitable, considering the reduced margin between cost and price, is essential but not trivial. For these reasons, a quantitative description of the microbiological system sustaining the process is pivotal: the development of mathematical models can support this description and predict possible improvements, which can be crucial for the viability of the biomethane sector.

Biogas production in anaerobic digestors generally follows four sequential phases carried out by the metabolism of the microbial consortium therein. Biopolymers such as carbohydrates are initially hydrolyzed into simple or simpler monomers and oligomers (e.g., sugars, stage I—hydrolysis), then fermented into alcohols, $CO_2$, volatile fatty acids (VFAs, e.g., propionate, butyrate, acetate), and $H_2$ (stage II—acidogenesis) [8]. These molecules are then converted into acetic acid (stage III—acetogenesis) and sequentially transformed into biomethane ($CH_4$) (stage IV—methanogenesis) (Figure 1) [8]. These reactions are carried out by different species of microbes, whose symbiosis and syntropy in the consortium are the key for the anaerobic digestor to function. In a simplified model of these reactions, described in Figure 1, stage I is carried out by glucose-fermenting acidogens, able to both hydrolyze the fibers and carbohydrates contained in the substrate and transform glucose into VFAs. Propionate and butyrate-degrading acetogens accumulate acetic acid from corresponding VFAs, whereas acetoclastic methanogens complete the reaction to biomethane [9–11].

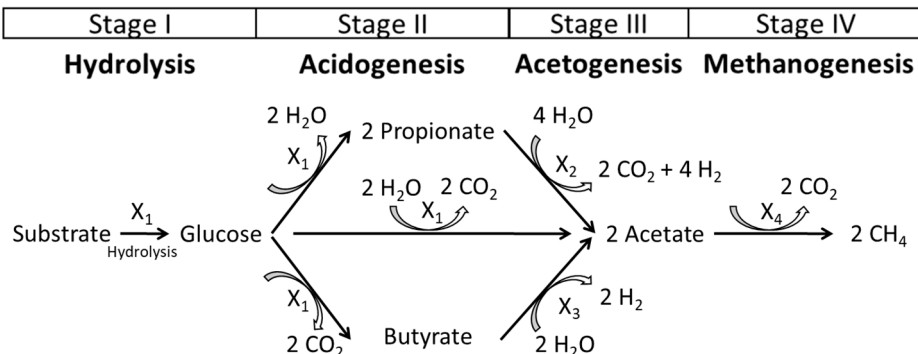

**Figure 1.** Simplified biochemical model of anaerobic digestion of biomass by the consortium of glucose-fermenting acidogens ($X_1$), propionate-degrading acetogens ($X_2$), butyrate-degrading acetogens ($X_3$), and acetoclastic methanogens ($X_4$).

From the process of the waste-derived biomethane, it is possible to create a mathematical model that is able to predict the behavior of the process itself in case specific parameters are modified, as for example in [10,11], where a simplified version of the Anaerobic Digestion Model No. 1 (ADM1) is described: this model focuses on the exploitation of activated sludges for wastewater treatments for biomethane production [9]. ADM1 has been also adjusted to stress the relevance of specific microbes, molecules, or operative steps [12–15], and applied to the use other residual biomasses, especially of lignocellulosic origin, such as olive mill solid wastes, maize silage, and vegetable crop residues [16–19].

The goal of the present work is to update this model with observations from the microbiological nature of the process, and to evaluate which industrially relevant parameters could impact the most on the final production of biomethane by the simulated microbial consortium.

## 2. Materials and Methods

### 2.1. Mathematical Model of Anaerobic Digestion

The model structure presented in [11] was chosen because it is a relatively new modeling approach and this model has been built with experimental data. The starting point of the work was the Anaerobic Digestion Model No. 1 (ADM1) developed by the International Water Association (IWA) for the use of activated sludge from the municipal wastewater treatment plants [9]. The ADM1 model has been successfully proposed for the production of biomethane from different residual sources [20,21]. Here, we use a simplified version of the ADM1 model, were hydrogenotrophic methanogens were omitted considering their lower impact on biomethane production if compared to acetoclastic methanogens [10]. In addition, only glucose was considered as carbon source, despite proteins are the most abundant nutrient in activated sludges [11]; indeed, the original ADM1 model describes glucose as the model monomer for acidogenesis in anaerobic digestion [9], as the aim was to use it for modeling biomethane production from biomasses enriched in carbohydrates, such as lignocellulosic ones. The equations taken into consideration modified from [11] were the following (the symbols used are explained in the next subsection):

$$\frac{dS_0}{dt} = -\beta \frac{S_0 X_1}{S_2 + S_3 + S_4 + K_{i,o}} + DY_e S_{oi} + \lambda \left(\sum_{i=1}^{6} b_i X_i\right) - DS_0 \tag{1}$$

$$\frac{dX_1}{dt} = (\mu_1 - b_1)X_1 - DX_1 \tag{2}$$

$$\frac{dS_1}{dt} = -Y_{glu/X1}\mu_1 X_1 + \beta \frac{S_0 X_1}{S_2 + S_3 + S_4 + K_{i,o}} - DS_1 + S_{1i} \tag{3}$$

$$\frac{dX_2}{dt} = (\mu_2 - b_2)X_2 - DX_2 \tag{4}$$

$$\frac{dS_2}{dt} = Y_{prop/X1}\mu_1 X_1 - Y_{prop/X2}\mu_2 X_2 - DS_2 + S_{2i} \tag{5}$$

$$\frac{dX_3}{dt} = (\mu_3 - b_3)X_3 - DX_3 \tag{6}$$

$$\frac{dS_3}{dt} = Y_{but/X1}\mu_1 X_1 - Y_{but/X3}\mu_3 X_3 - DS_3 + S_{3i} \tag{7}$$

$$\frac{dX_4}{dt} = (\mu_4 - b_4)X_4 - DX_4 \tag{8}$$

$$\frac{dS_4}{dt} = Y_{acet/X1}\mu_1 X_1 + Y_{acet/X2}\mu_2 X_2 + Y_{acet/X3}\mu_3 X_3 - Y_{acet/X4}\mu_4 X_4 - DS_4 + S_{4i} \tag{9}$$

$$Q = Y_{CH4/X4}\mu_4 X_4 \tag{10}$$

### 2.2. Variables and Constant Description

The variables and constants considered in this work were obtained from [11], which described anaerobic digestion in a continuously stirred tank reactor.

#### 2.2.1. Variables

$Q$ (L/d) is the biogas yield over time, $S_0$ (g/L) is the concentration of soluble organic compounds, measured as volatile solids, $S_{NH4^+}$ is the concentration of ammonia, $S_1$ (g/L) is the concentration of glucose, $S_2$ (g/L) is the concentration of propionate, $S_3$ (g/L) is the concentration of butyrate, $S_4$ (g/L) is the concentration of acetate, $X_1$ (g/L) is concentration of glucose-fermenting acidogens, $X_2$ (g/L) is concentration of propionate-degrading acetogens, $X_3$ (g/L) is concentration of butyrate-degrading acetogens, $X_4$ (g/L)

is concentration of acetoclastic methanogens, and $\mu_1$, $\mu_2$, $\mu_3$, and $\mu_4$ (d$^{-1}$) refer to specific growth rates, described as follows:

$$\mu_1 = \frac{\mu_{max1} S_1}{K_{S1} + S_1} \tag{11}$$

$$\mu_2 = \frac{\mu_{max2}}{(1 + K_{S2}/S_2)(1 + S_4/K_{i,acet/prop})} \tag{12}$$

$$\mu_3 = \frac{\mu_{max3}}{(1 + K_{S3}/S_3)(1 + S_4/K_{i,acet/but})} \tag{13}$$

$$\mu_4 = \frac{\mu_{max4} K_{i,NH4+} S_4}{(K_m X_4 + S_4)(K_{i,NH4+} + S_{NH4+})} \tag{14}$$

### 2.2.2. Constants

There are several constants in the equations and their values are as follows: $D$ (d$^{-1}$) = 0.1072—dilution rate; $\beta$ (d$^{-1}$) = 0.31—hydrolytic rate; $K_{i,o}$ (g/L) = 0.23—inhibition constant, reflecting the decrease of hydrolytic rate due to VFAs accumulation; $K_{i,NH4^+}$ (g/L) = 0.5 —inhibition constant reflecting the decrease of acetoclastic methanogenesis rate due to ammonia accumulation; $K_{i,acet/prop}$ (g/L) = 0.96—product inhibition constant, reflecting the decrease of propionate degradation rate due to acetate accumulation; $K_{i,acet/but}$ (g/L) = 0.72— product inhibition constant, reflecting the decrease of butyrate degradation rate due to acetate accumulation; $Y_e$ = 0.55—coefficient of decomposition, counting what part of insoluble organic compounds are transformed to soluble compounds; $S_{oi}$ = 30.6 g/L— concentration of insoluble organic compounds, measured as total solids; $S_{i1}$, $S_{i2}$, $S_{i3}$, and $S_{i4}$ (g/L) are the concentrations of the corresponding substrates in the influent; $S_{i1}$ = 5.1, $S_{i2}$ = 1.6, $S_{i3}$ = 0.1, and $S_{i4}$ = 3.1.

$K_m$ = 1.3—coefficient in the Contois growth rate model for $\mu_4$, reflecting the decrease of acetoclastic methanogenesis rate due to biomass accumulation; $K_{S1}$ (g/L) = 4.8—saturation constant for glucose-fermenting acidogens; $K_{S2}$ (g/L) = 0.93—saturation constant for propionate-degrading acetogens; $K_{S3}$ (g/L) = 0.176—saturation constant for butyrate-degrading acetogens

$\mu_{max1}$ (d$^{-1}$) = 0.7—maximum specific growth rate of glucose-fermenting acidogens at 34 °C; $\mu_{max2}$ (d$^{-1}$) = 0.54—maximum specific growth rate of propionate-degrading acetogens at 34 °C; $\mu_{max3}$ (d$^{-1}$) = 0.68—maximum specific growth rate of butyrate-degrading acetogens at 34 °C; $\mu_{max4}$ (d$^{-1}$) = 0.45—maximum specific growth rate of acetoclastic methanogens at 34 °C; $b_i$ ($i$ = 1, ..., 4)—mortality rates for each of the four microbial populations (it was supposed that $b_i = 0.05\mu_{maxi}$). It was assumed that a part of the dead cells is transformed into soluble organics with recycling conversion factor $\lambda$ ($\lambda > 0$ and $\lambda < b_i$).

Yield coefficients of production or consumption: $Y_{glu/X1}$ = 12.9 g/g biomass, $Y_{acet/X1}$ = 20 g/g biomass, $Y_{prop/X1}$ = 2.94 g/g biomass, $Y_{prop/X2}$ = 10.2 g/g biomass, $Y_{but/X1}$ = 3.08 g/g biomass, $Y_{but/X3}$ = 11.9 g/g biomass, $Y_{acet/X2}$ = 8 g/g biomass, $Y_{acet/X3}$ = 1.54 g/g biomass, $Y_{acet/X4}$ = 16 g/g biomass, and $Y_{CH4/X4}$ = 4 L/g biomass.

### 2.3. Balancing the Reaction Equations of Anaerobic Digestion Mathematical Model

The biochemical reactions from glucose to biomethane described in the model were stoichiometrically balanced and analyzed by using the biochemical thermodynamics calculator eQuilibrator [22]. For each equation, the estimated Gibbs free energy ($\Delta_r G'^\circ$) and equilibrium constant (K'$_{eq}$) were calculated. The values of pH, pMg, and ionic strength were kept as the default ones (7.5, 3.0, and 0.25 M, respectively).

### 2.4. Simulation Studies

The mathematical model was recreated on Simulink (Academic use version, Math-Works, Natick, MA, USA) and run via data uploaded on MATLAB$^{®}$ (R2019b version,

MathWorks, Natick, MA, USA). To improve the model, different simulation scenarios were run to validate model behavior. Based on our observations, the differential equations of the model were modified on Simulink. Simulations were then run again to verify appropriate behavior. To identify significant process parameters, the values for some parameters were modified. Thus, we can foresee the effect of such modifications on the production of biomethane. Excel (Office 365, Microsoft, Albuquerque, NM, USA) was then used to calculate equations of the relationship between single parameters ($D$, $Y_{acet/X1}$, $S_{1i}$, $S_{0i}$, $Y_e$) and biomethane production ($Q$), described in Section 3.4.

## 3. Results and Discussion

### 3.1. Biochemical Description of the Model

Microbial metabolism reflects microbial biodiversity: different species often display different abilities in utilizing the same carbon source. Because of the heterogeneous nature of microbial consortia, understanding the main reactions occurring within is of utmost industrial interest [8]. As mentioned, the production of biomethane from biogas is composed of four main phases that, although they are sequential to each other, follow different metabolic branches. In fact, as shown in Figure 1, different routes can lead to the accumulation of acetate, which are then transformed into biomethane by acetoclastic methanogens. A direct metabolism from glucose to acetate is in fact combined by the alternative production of butyrate or propionate, later converted into acetate as well. Despite the presence of other carbon sources rather than glucose, it is indicated in the ADM1 as the model monomer for acidogenesis in anaerobic digestion [9]. The three metabolisms could be described stoichiometrically with the chemical equations enlisted in Table 1.

**Table 1.** Stoichiometric description of the reactions occurring during anaerobic digestion from glucose to acetate, via propionate (1) or butyrate (2) production, or direct fermentation to acetate (3), with kinetic parameters of such reactions.

| Reaction | Estimated $\Delta_r G'^\circ$ (KJ/Mol) | $K'_{eq}$ |
|---|---|---|
| (1) Glucose + 2 $H_2O$ <=> 4 $CO_2$ + 2 $CH_4$ + 4 $H_2$ | $-148.6 \pm 35.0$ | $1.1 \times 10^{26}$ |
| (2) Glucose + 2 $NAD^+$ + 2 $H_2O$ <=> 4 $CO_2$ + 2 $CH_4$ + 2 $H_2$ + 2 NADH | $-217.7 \pm 28.6$ | $1.5 \times 10^{38}$ |
| (3) Glucose + 4 $NAD^+$ + 2 $H_2O$ <=> 4 $CO_2$ + 2 $CH_4$ + 4 NADH | $-286.9 \pm 26.3$ | $1.9 \times 10^{50}$ |

The analysis of these equations makes clear that with the direct fermentation of glucose into acetate, $NAD^+$ becomes the electron sink of the reaction (Reaction 3, Table 1), whereas $H_2$ is the receiver of the four electrons (Reaction 1, Table 1) when propionate-degrading acetogens are involved. Accordingly, butyrate as mid-step involves the use of both $H_2$ and $NAD^+$ as electron sinks of the fermentation (Reaction 2, Table 1). Considering kinetic parameters such as the estimated Gibbs free energy ($\Delta_r G'^\circ$) and equilibrium constant ($K'_{eq}$), it was clear that the direct fermentation of glucose into acetate is thermodynamically the most favorable, suggesting that the microorganisms involved in such metabolism are pivotal for the accumulation of biomethane. In agreement with this observation, bioaugmentation of acetate-type fermentation species have been proposed to ameliorate anaerobic digestion of residual biomasses into biomethane [23].

### 3.2. Microbiological Analysis of ADM1 Model

The differential equations describing the anaerobic digestion into biomethane from [11], developed for the use of activated sludge from the municipal wastewater treatment plants, were recreated on Simulink in a simplified version, by omitting hydrogenotrophic methanogens, since their yield of methane was calculated to be inferior to acetoclastic methanogens (0.8 and 4 g/g biomass, respectively [11]). In fact, acetoclastic methanogens

are responsible for most of the biomethane produced during anaerobic digestion (up to 70% of the total) [10]. Given the considerations arising from the thermodynamical analysis of the reactions leading to biomethane, we decided to use the model to quantify the influence of the metabolism of the fermenting microbes ($X_1$, $X_2$, and $X_3$) on the production of biomethane. We singularly simulated the nullification of acetate production yields by $X_1$, $X_2$, and $X_3$ ($Y_{acet/X1}$, $Y_{acet/X2}$, and $Y_{acet/X3}$, respectively) to assess their impact on the final biomethane yield over time ($Q$): the lower this value when a single parameter is set to zero, the higher the importance of such element in the system. When $Y_{acet/X2}$ and $Y_{acet/X3}$ were set to zero, $Q$ did not decrease significantly from the original one ($Q$ = 0.45 L/d, Figure 2A), whereas $Y_{acet/X1}$ = 0 resulted in $Q$ = 0.18 L/d, witnessing the importance of the metabolism of $X_1$ on the overall process over $X_2$ and $X_3$ (in accordance with the thermodynamical analysis). We then expanded the simulation by setting to zero the yield of glucose consumption by $X_1$ ($Y_{glu/X1}$ = 0) to completely eliminate the action of this species earlier in the process, expecting a strong reduction of $Q$. Surprisingly, the simulation resulted in an infinite production of biomethane in the first days of fermentation (Figure 2B), underlying some inaccuracy in the model itself. In the model $X_1$, both hydrolyzed fibers and fermented glucose into VFAs; therefore, the subsequent production of biomethane is strongly dependent on its activity. Since $Y_{glu/X1}$ represents the ability of $X_1$ to consume glucose and, therefore, to grow and produce VFAs, the fact that its simulated nullification was so beneficial for the production of biomethane was at least suspicious from a microbiological point of view. Therefore, we analyzed the description of the equations to understand the reasons of this unexpected finding and how to solve it, for better representing the microbiological reality of anaerobic digestion and for further improving the model itself.

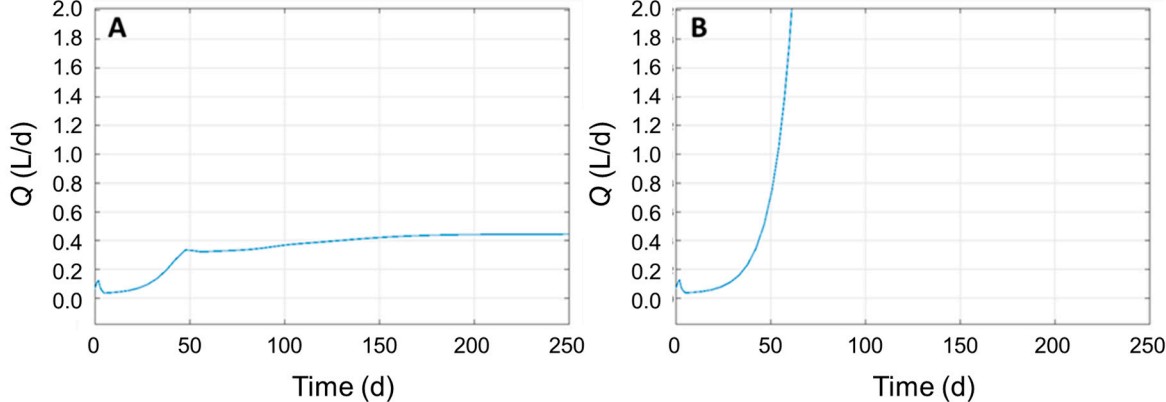

**Figure 2.** Simulation of biomethane yield ($Q$) during time using parameters from [11] (panel **A**) and with $Y_{glu/X1}$ = 0 (panel **B**).

### 3.3. Update to ADM1 Model Structure

Analyzing the equations of the model indicated the mathematical reason for such behavior. $Y_{glu/X1}$ appears in Equation (3) as a negative contributor to the titer of glucose in the digestor. The nullification of $Y_{glu/X1}$, therefore, increased the amount of glucose, which in turn increased $\mu_1$ (Equation (11)) and the titer of $X_1$ (Equation (1)), and subsequently, the titer of the microbial species as well. The paradox laid on the fact that although $X_1$ was simulated of not being able to consume glucose, it was still growing and producing VFAs, while microbiologically the opposite should occur. In order to meet this fundamental requirement, we modified the equations describing microbial growth (Equations (2), (4), (6) and (8)) by adding the corresponding yields of substrate consumption ($Y_{glu/X1}$, $Y_{prop/X2}$, $Y_{but/X3}$, $Y_{acet/X4}$) as follows:

$$\frac{dX_1}{dt} = Y_{glu/X1}(\mu_1 - b_1)X_1 - DX_1 \tag{15}$$

$$\frac{dX_2}{dt} = Y_{prop/X2}(\mu_2 - b_2)X_2 - DX_2 \tag{16}$$

$$\frac{dX_3}{dt} = Y_{but/X3}(\mu_3 - b_3)X_3 - DX_3 \tag{17}$$

$$\frac{dX_4}{dt} = Y_{acet/X4}(\mu_4 - b_4)X_4 - DX_4 \tag{18}$$

With the updated equations, the original parameter values permitted a simulation with $Q$ = 0.9698 L/d, whereas the nullification of $Y_{glu/X1}$ reduced the value to $Q$ = 0.1104 L/d, in accordance with the modifications proposed for the model (Figure 3). The value of $Q$ was not zero because of the acetate present in the influent ($S_{i4}$), transformed by $X_4$ into biomethane according to Equations (9) and (10). Furthermore, the value of $Q$ obtained with this new version of the model was higher compared to the original [11], although it should be noticed that the constant values for updated model need to be validated experimentally. Nevertheless, the goal of this work was to provide an updated version of the model that could be a closer description of the microbial consortium responsible for the anaerobic digestion. Indeed, despite that the model analyzed here does not involve hydrogenotrophic methanogens, since their minor contribution in biomethane production compared to acetoclastic methanogens [10], the proposed modifications can be considered valid for the original ADM1 model as well.

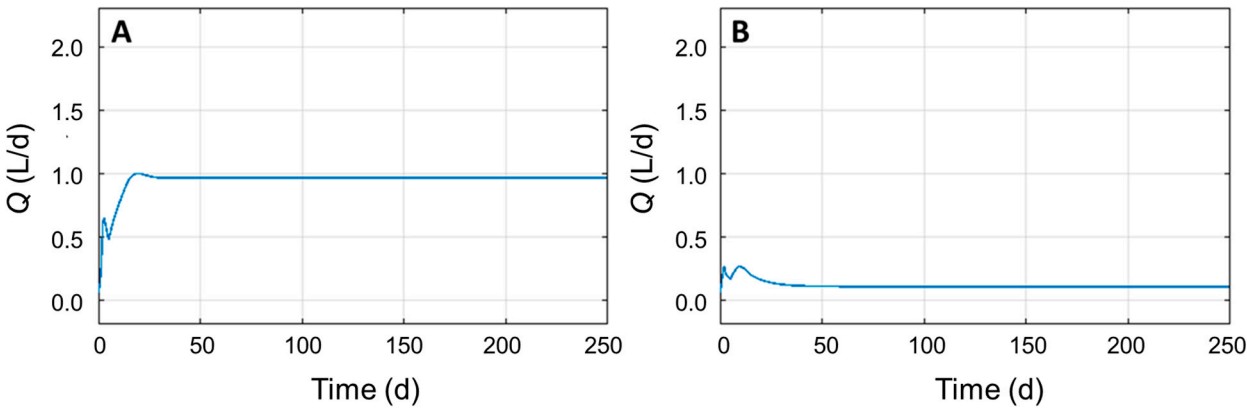

**Figure 3.** Simulation of biomethane yield ($Q$) over time with updated equations using parameters from [11] (panel **A**) and with $Y_{glu/X1} = 0$ (panel **B**).

### 3.4. Identifying Significant Process Parameters

After updating the model by introducing the yields of substrate consumption in the equations, it was possible to simulate the growth of the various microbial species present in the model ($X_1$, $X_2$, $X_3$, $X_4$). As a starting point, the dilution rate ($D$) was modified according to the maximum value available at the plant of an industrial partner of the project ($D$ = 0.14 d$^{-1}$), with 5000 L as the operative volume. The updated set of values resulted in $Q$ = 1.198 L/d, displayed in Figure 4 together with other parameters, such as substrate, glucose and VFAs titer, microbial titer, and specific growth rate. The stoppage time was set to 60 d, since the maximum $Q$ was already reached.

The production of biomethane ($Q$) as a function of different parameters in the model was identified based on the simulation results. The interaction between the dilution rate ($D$) and $Q$ can be described by a second-order polynomial ($Q = -9.9166D^2 + 8.8125D + 0.124$, R$^2$ = 0.99), underlying the limit of the dilution that can cause wash out. With the polynomial above, it was also possible to calculate the dilution rate that provided the highest production of biomethane ($D$ = 0.4458 d$^{-1}$, $Q$ = 2.062 L/d). The ability to foresee this value is clearly an advantage of the use of mathematical modeling.

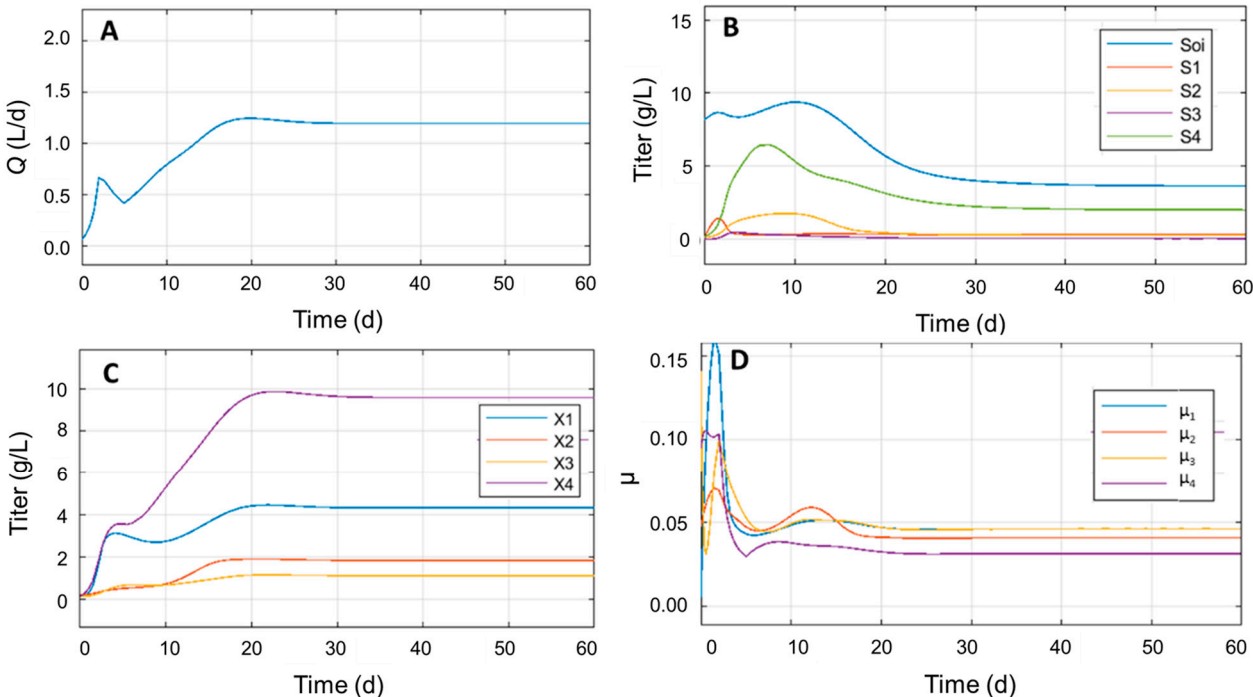

**Figure 4.** Simulation of biomethane yield over time (panel **A**), compound titers (panel **B**), microbial titers (panel **C**), and microbial specific growth rates (panel **D**) over time with updated equations, and $D$ = 0.14 $\text{d}^{-1}$.

$Q$ was found to be a linear function of some parameters of the model (listed in Table 2), since their mathematical relationships could be described with a first-order polynomial. Nevertheless, the slope was not sufficient for comparisons, since the parameters have different orders of magnitudes among themselves. Therefore, we decided to calculate ratios between the predicted biomethane yields over time and the studied parameter values as denominator. Table 2 shows the simulation results and illustrates the calculations of the ratios. The last column in Table 2 shows the ratio that is used to understand how much an increase in the parameter value affects the increase of biomethane production.

**Table 2.** Effect of the increment of the value of some parameters of the anaerobic digestion model on the value of biomethane production ($Q$).

| Parameter | Value | Ratio | $Q$ Value | $Q$ Ratio | $Q$ Ratio/Parameter Value Ratio |
|---|---|---|---|---|---|
| $Y_{acet/X1}$ | 20 | | 1.198 | | |
| | 40 | 2 | 1.933 | 1.62 | 0.81 |
| | 60 | 3 | 2.548 | 2.13 | 0.71 |
| $S_{1i}$ (g/L) | 5.1 | | 1.198 | | |
| | 10.2 | 2 | 1.504 | 1.26 | 0.63 |
| | 15.3 | 3 | 1.807 | 1.51 | 0.50 |
| $S_{0i}$ (g/L) | 30.6 | | 1.198 | | |
| | 61.2 | 2 | 2.001 | 1.67 | 0.84 |
| | 91.8 | 3 | 2.796 | 2.33 | 0.78 |
| $Y_e$ | 0.55 | | 1.198 | | |
| | 1.1 | 2 | 2.001 | 1.67 | 0.84 |
| | 1.65 | 3 | 2.796 | 2.33 | 0.78 |

Giving the importance of $X_1$ metabolism in anaerobic digestion, in particular for the production of acetate, $Y_{acet/X1}$ was modified to assess its impact on $Q$. Similarly, values for the initial concentration of glucose in the influent ($S_{1i}$) were considered, to simulate the effect of the addition of another residual biomass containing glucose. Finally, the

concentration of insoluble organic compounds ($S_{oi}$), and the coefficient of decomposition ($Y_e$), counting which part of the insoluble organic compounds are transformed to soluble compounds, were considered as well. Table 2 shows that none of the modified parameters caused the same increase in $Q$ and that the more the value was increased the lower was the effect on the corresponding ratio. In addition, implementation of $S_{1i}$ had only minor effects, showing that additional glucose would not greatly impact $Q$. On the other hand, $S_{oi}$ and $Y_e$ showed to be crucial for a relevant implementation of $Q$; unsurprisingly, the same increment of $S_{oi}$ and $Y_e$ produced the same values of $Q$, since they are both factors of Equation (1).

Nevertheless, increasing the initial concentration of insoluble organic compounds could be problematic from a technical point of view; furthermore, organic loading rates exceeding the decomposition or the hydrolysis rates can determine a decline in methane production [5,20,24]. On the other hand, the coefficient of decomposition is a parameter that could be targeted more easily, as it can be modified by pretreating the biomass with enzymes. Although the possibility to exploit enzymatic hydrolysis in increasing biomethane potential in residual biomasses was already explored in [25–27], this work fosters the importance to predict the impact of such a component by computational analysis. This concept is relevant in a scientific and industrial context where the biochemical methane potential (BMP) tests, widely applied to analyze anaerobic digestion processes, still lack standardization [28]. Since the use of enzymes remains a potential threat to the economic sustainability of the process considering the cascading principles [29], a techno-economic analysis would be needed for any single case: computational modeling is, therefore, useful, as it can project the impact of different enzyme loading on the improvement of methane production. It is worth mentioning here that an available alternative is the in situ production of such enzymes: in a recent example, the organic fraction of municipal solid waste was used for triggering the secretion of enzymes by *Aspergillus niger* and promoting the subsequent anaerobic digestion by microbial consortia, whose biomethane potential increased [30].

Furthermore, when $Q$ was considered as a function of the hydrolytic rate ($\beta$), a logarithmic equation was obtained ($Q = 0.1609ln(\beta) + 1.3021$, $R^2 = 0.91$), showing that an improvement in the ability of the microbial consortium to hydrolyze fibers is not proportionally beneficial in the production of biomethane (Figure S1). These observations from the simulation of anaerobic digestion of sludge may pave the way for further implementation of the real industrial process, and in turn, to the validation of the modified equations from the original ADM1.

### 4. Practical Applications of This Work and Future Research

In this work, the ADM1 model was updated to comprise substrate consumption yields in the equations, describing more logically microbial growth during anaerobic digestion. The production of biomethane is a direct consequence of primary metabolism and, therefore, of biomass growth, which can occur only if substrate is consumed. Furthermore, we identified the relationships between the production of biomethane and some specific process parameters, pinpointing and predicting that organic loading can be decisive to improve the final production: we are aware that the practical implementation of this prediction is not trivial. Since the coefficient of decomposition impacts biomethane production as well, the use of enzymes can be envisaged as a direct approach of improvement, to be further evaluated from an economical point of view. Future analysis will involve the validation of the new model equations with the values obtained by real experiments in an industrial plant.

### 5. Conclusions

The quest for more sustainable energy sources is intertwined with the development of technologies able to valorize renewable biomasses, with increased appeal if residual ones are used. Among those, sludge from wastewater treatment and lignocellulosic material are

among the most promising, because of their abundance, low cost, and intrinsic potential to be fermented into biogas. Since the complex nature of a process also depends on heterogenous microbial consortia, the deployment of mathematical modeling of anaerobic digestion can be a concrete help to direct possible manipulations. In this work, we showed that the thermodynamical and microbiological analysis of ADM1 model can lead to its improvement in simulating the anaerobic digestion process to obtain biomethane.

**Supplementary Materials:** The following are available online at https://www.mdpi.com/article/10.3390/fermentation7040237/s1, Figure S1: Biomethane production ($Q$) as function of hydrolytic rate ($\beta$).

**Author Contributions:** Conceptualization, S.B., M.R. and A.S. (Aki Sorsa); methodology, S.B., M.R. and A.S. (Aki Sorsa); validation, S.B., M.R. and A.S. (Aki Sorsa); formal analysis, S.B.; investigation, S.B.; resources, M.R., A.S. (Anu Sirviö) and P.B.; data curation, S.B.; writing—original draft preparation, S.B.; writing—review and editing, S.B., M.R., A.S. (Aki Sorsa), A.S. (Anu Sirviö) and P.B.; supervision, M.R., A.S. (Aki Sorsa) and P.B.; project administration, A.S. (Anu Sirviö) and P.B.; funding acquisition, A.S. (Anu Sirviö) and P.B. All authors have read and agreed to the published version of the manuscript.

**Funding:** This research project has been supported by the European Institution of Innovation & Technology (EIT) KIC-RawMaterials project ADMA2 (practical training between Academia and Industry during doctoral studies) funding number 18252.

**Institutional Review Board Statement:** Not applicable.

**Informed Consent Statement:** Not applicable.

**Data Availability Statement:** Not applicable.

**Conflicts of Interest:** The authors declare no conflict of interest.

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
