# Peer review of "Mathematical Analysis and Update of ADM1 Model for Biomethane Production by Anaerobic Digestion"

_fermentation, doi:10.3390/fermentation7040237_

Round 1

Reviewer 1 Report

The researchers aimed to modify the Anaerobic Digestion Model No. 1 (ADM-1), describing the biomethane production from activated sludge and to use the model for identifying relevant parameters to improve biomethane production. ADM-1 is a model used for simulation and in this study, there was not an experimental procedure to support the results. In the introduction of this study a big part should be more focused in ADM-1 (previous studies) as it’s alteration to better describe the anaerobic biological process was the main objective. In the Materials and Methods section a lot of information is not described well enough (i.e. what were the values of the parameters you used in the eQuilibrator, for pH etc.) as well as the purpose of using specific tools is missing (i.e. Excel was used for which calculation of equations?). In the Results section you omitted the hydrogenotrophic methanogens in your ADM-1 altered model as also some other parameters. These changes cannot occur in real conditions at an anaerobic reactor and obviously they cannot be related with the use of the ADM-1 especially in commercially scale processes that they were mentioned in the manuscript. Furthermore, you point out that the coefficient of decomposition is a parameter that could be targeted more easily as it can be modified by pretreating the biomass with enzymes or composting. The use of enzymes in large-scale processes is usually unprofitable and composting will probably lower significantly the biomethane potential of the substrates for the anaerobic process (discussion as also references are missing). In the section of References after the number 14 the bibliography is missing in the current manuscript.

Author Response

The researchers aimed to modify the Anaerobic Digestion Model No. 1 (ADM-1), describing the biomethane production from activated sludge and to use the model for identifying relevant parameters to improve biomethane production. ADM-1 is a model used for simulation and in this study, there was not an experimental procedure to support the results.

In the introduction of this study a big part should be more focused in ADM-1 (previous studies) as it’s alteration to better describe the anaerobic biological process was the main objective.

We added in the introduction some insights and references in the use of ADM-1 modified in terms of biomasses considered or equations to better match specific parameters to be analyzed (L81-85)

In the Materials and Methods section a lot of information is not described well enough (i.e. what were the values of the parameters you used in the eQuilibrator, for pH etc.) as well as the purpose of using specific tools is missing (i.e. Excel was used for which calculation of equations?).

We added information regarding pH, pMG and ionic strength values used at L155-6. Moreover, we explicated the use of excel at L66-7

In the Results section you omitted the hydrogenotrophic methanogens in your ADM-1 altered model as also some other parameters. These changes cannot occur in real conditions at an anaerobic reactor and obviously they cannot be related with the use of the ADM-1 especially in commercially scale processes that they were mentioned in the manuscript.

The reviewer correctly underlines the omission of hydrogenotrophic methanogens in the ADM-1 model, as mentioned from L206. In that part of the manuscript we also stressed that acetoclastic methanogens are responsible for up to 70% of the total biomethane produced during anaerobic digestion (L209-10). We therefore deployed a simplified version of ADM-1 model, bearing in mind that the modifications to the equations of the original model and the analysis on specific parameters would be anyhow valid. Indeed, the nullification of glucose consumption by X1 (Yglu/X1) would simulate an infinite production of biomethane even if considering hydrogenotrophic methanogens, as the equations we considered for the simplified version of ADM-1 cover the first reactions of the overall pathway, from sugars to biomethane. Therefore, they will affect everything happening downstream. We added at L96-9 (materials & methods section) and L260-63 (results section) a sentence to stress this concept and to inform the reader about our assumptions.

Furthermore, you point out that the coefficient of decomposition is a parameter that could be targeted more easily as it can be modified by pretreating the biomass with enzymes or composting. The use of enzymes in large-scale processes is usually unprofitable and composting will probably lower significantly the biomethane potential of the substrates for the anaerobic process (discussion as also references are missing).

We thank the reviewer for highlighting the importance to further stress this part of the manuscript. First of all, we erased reference to composting, because as the reviewer mentioned it might not be compatible with anaerobic digestion, because of the different environmental conditions needed (aerobic or anaerobic). Regarding enzymatic hydrolysis, its application is clearly adding to the cost of a bioprocess, particularly when dealing with a cheap product such as biomethane. Therefore, mathematical modeling and computational simulation could be really crucial to understand the impact of enzymatic hydrolysis on the final biomethane production and yield on time, and by predicting how to optimize the enzyme loading could indicate the best compromise in terms of costs. We added both in the results and in the discussion section a specific part dedicated to enzymatic hydrolysis (L326-339).

In the section of References after the number 14 the bibliography is missing in the current manuscript.

We really thank the reviewer for pointing this out, it looks like it was a problem of the program used for the bibliography. We now fixed it and the bibliography is complete.

Reviewer 2 Report

The manuscript titled “Mathematical analysis and update of ADM-1 model for biomethane production by anaerobic digestion” has two aims:
 The fist is to modify the Anaerobic Digestion Model No. 1 (ADM-1), describing biomethane production from activated sludge, matching with its actual microbial nature
The second is to modify the relevant parameters to improve biomethane production.
The goal of the manuscript is to update the ADM1 with observations from the microbiological nature of the process
Finally, the new updated model was tested by modifying several parameters and the coefficient of decomposition was identified to increase biomethane production.

1. The authors in the paragraph 3.1 put their attention in the metabolism of glucose and in table 1 they reported the stoichiometric description of the reactions occurring during anaerobic digestion from glucose to acetate; moreover, table1 shows the estimated Gibbs free Energy and the equilibrium constant. It is not clear why they focus their attention in glucose transformation to acetate since the activated sludge is mainly composed of proteins.
The authors have to clarify better this choice.

2. Paragraph 3.2 it is not clear. Authors mast describe in more in depth their work

2. In paragraphs 3.4 and 4 the authors highlight that the coefficient of decomposition impacts on biomethane production. I believe that it is well known since the beginning of Anaerobic Digestion process study.
Are you sure that is it a novelty?
In the well-known ADM1 both the disintegration and hydrolysis processes are modelled as a fist order kinetic processes and they are the only processes were microorganisms are not directly involved. Therefore, if the coefficient of decomposition is so important why authors focus their attention on the glucose reaction occurring during anaerobic digestion.  
Probably rewriting paragraph 3.2 the conclusions will  be more clear

Author Response

The manuscript titled “Mathematical analysis and update of ADM-1 model for biomethane production by anaerobic digestion” has two aims: The first is to modify the Anaerobic Digestion Model No. 1 (ADM-1), describing biomethane production from activated sludge, matching with its actual microbial nature the second is to modify the relevant parameters to improve biomethane production. The goal of the manuscript is to update the ADM1 with observations from the microbiological nature of the process. Finally, the new updated model was tested by modifying several parameters and the coefficient of decomposition was identified to increase biomethane production.

The authors in the paragraph 3.1 put their attention in the metabolism of glucose and in table 1 they reported the stoichiometric description of the reactions occurring during anaerobic digestion from glucose to acetate; moreover, table1 shows the estimated Gibbs free Energy and the equilibrium constant. It is not clear why they focus their attention in glucose transformation to acetate since the activated sludge is mainly composed of proteins.
The authors have to clarify better this choice.

The estimation of Gibbs free energy and equilibrium constants of the reactions leading to biomethane production were considered starting from glucose as substrate, mainly to resemble the simulation with ADM-1 performed by Simeonov et al. (2012). In that work, the authors measured the amount of both sugars and proteins in the activated sludge, confirming the argument of the reviewer that this biomass contains more of the latter of the two: however, they did not disclose the real reasons of this choice. Nevertheless, in the original ADM-1 model by the IWA (Batstone et al. 2002) the task group described glucose as the model monomer for acidogenesis (acetate, propionate, butyrate). Therefore, we decided to follow the equations and parameters publicly described by Simeonov et al. (2012), that involve a simplified version of ADM-1 (starting from glucose), yet useful to analyze anaerobic digestion from the modelling point of view, of both activated sludge and other biomasses of lignocellulosic origin, whose main component is often cellulose. In order to clarify this concept, we added a sentence in the materials and methods section at L99-102 and in the results section L178-80 as well. We also stressed in the introduction several examples of application of ADM-1 model to the use of lignocellulosic biomasses L82-4.

Paragraph 3.2 it is not clear. Authors mast describe in more in depth their work

We expanded the concepts explained in section 3.2 in order to highlight the goals of this part of the work.

In paragraphs 3.4 and 4 the authors highlight that the coefficient of decomposition impacts on biomethane production. I believe that it is well known since the beginning of Anaerobic Digestion process study.
Are you sure that is it a novelty?

The use of enzymatic hydrolysis as form of pre-treatment of the biomass to be used in anaerobic digestion was already explored, as in the references that we added in L326-28. Nevertheless, because of the lack of standardized analysis for biomethane potential, mathematical modeling and computational simulation represent powerful tools to be exploited. The fact that the coefficient of decomposition was identified in this work as a crucial parameter to improve biomethane production is an indication of the reliability of the proposed modifications to ADM-1. Furthermore, the impact on the overall costs of enzymes could be forecasted by the simulation, by understanding the interdependence between the use of enzymatic hydrolysis, the biomass decomposition and the biomethane yield on time. We stressed this issue in L329-35.

In the well-known ADM1 both the disintegration and hydrolysis processes are modelled as a fist order kinetic processes and they are the only processes were microorganisms are not directly involved. Therefore, if the coefficient of decomposition is so important why authors focus their attention on the glucose reaction occurring during anaerobic digestion.

In the ADM-1 model two steps from substrate to monosaccharides are described. Neither of these steps involve microbes and follow first order kinetics in ADM-1, as the reviewer states. However, hydrolysis reactions include microbial activity which is considered in our study as well as in (Simeonov 2012). We study anaerobic digestion with glucose as a monomer to evaluate the process with other biomasses such as lignocellulosic ones, by considering the whole reaction chain from substrate to biomethane. Furthermore, our simulation results showed that the coefficient of decomposition is an important parameter to be addressed to improve biomethane production. We believe that with the findings described in our paper, the ADM-1 model can be predictive in different application areas and can also provide indications for further studies and process development.

Round 2

Reviewer 1 Report

It is ok for publication.

Author Response

We really thank the reviewer for the fruitful suggestions that provided hints to further improve the manuscript.

Reviewer 2 Report

Authors have improved their work, in the present form the paper can be published.

Author Response

(The authors gave the same response as above.)
